# Marine Sources of DHA-Rich Phospholipids with Anti-Alzheimer Effect

**DOI:** 10.3390/md20110662

**Published:** 2022-10-25

**Authors:** Inês Ferreira, Amélia P. Rauter, Narcisa M. Bandarra

**Affiliations:** 1Centro de Química Estrutural, Institute of Molecular Sciences, Faculdade de Ciências, Universidade de Lisboa, 1749-016 Lisboa, Portugal; 2Division of Aquaculture, Upgrading and Bioprospecting, Portuguese Institute of the Sea and Atmosphere, 1495-165 Lisboa, Portugal; 3CIIMAR, Interdisciplinary Centre of Marine and Environmental Research, University of Porto, 4050-123 Porto, Portugal

**Keywords:** Alzheimer’s disease, phospholipids, docosahexaenoic acid, marine sources

## Abstract

Alzheimer’s disease (AD) is a complex and progressive disease, which affects millions of people around the world. Despite the many efforts over the years to find efficient therapeutics, there is no cure yet. Nonetheless, many compounds have been proven to decrease Alzheimer’s symptoms. After a short overview of the hypotheses considered in AD drug development and the drugs approved for AD treatment, which lead to symptom release, we focus on the valorization of natural marine sources that decrease AD symptoms, particularly on docosahexaenoic acid (DHA), an important component in membrane phospholipids and the most abundant n−3 polyunsaturated fatty acids (PUFA) found in gray matter of the brain and in retina and on the DHA-containing phospholipids (DHA-PLs) present in marine sources, namely fish, krill, mollusks and in fisheries and aquaculture by-products. DHA-PLs’ bioactivities are presented, namely their properties in anti-neurodegeneration, neuroinflammation, as anticancer agents, as well as their benefits to obesity and visual problems. Fisheries and aquaculture by-products are also highlighted as they have a high content of DHA and DHA-rich phospholipids, can be extracted by green methodologies and should be considered in a circular economy for a healthy sustainable future.

## 1. Neurodegenerative Diseases

Neurodegenerative diseases are defined by a deterioration in cognitive function due to a continuous loss of neurons or the gathering of misfolded proteins in different areas of the central nervous system (CNS), which leads to brain atrophy, causing several cognitive, psychiatric and motor complications [1,2,3]. Due to the accelerated rate of the elderly population worldwide and the increasing life expectancy, the World Health Organization (WHO) expects that in the current century, neurodegenerative diseases will be the highest concern in health issues [4]. Mental illnesses, nowadays, affect about 1 billion people around the world. It is also known that dementia, a characteristic of many neurodegenerative diseases, is the seventh-leading cause of death in the world.

The most common neurodegenerative diseases are Alzheimer´s disease (AD), Parkinson´s disease (PD), amyotrophic lateral sclerosis (ALS) and Huntington’s disease (HD) [5].

Beyond ageing, genetic changes combined with environmental factors contribute to neurodegenerative diseases. This type of disease has three stages. The early stage, middle stage where the symptoms become clearer and the late stage where the patients are near to total dependence and inactivity.

Neurodegenerative diseases normally begin with inflammatory disorders. Inflammation processes have a protective purpose, but when dysregulation occurs, it triggers a cascade of events, initiating an immune response, leading to cell death [4].

Neurodegeneration occurs mainly in the later stages of life and implies mechanisms, such as oxidative stress, calcium dysregulation, mitochondrial dysfunction, neuroinflammation, among others. Regarding this, different neurodegenerative diseases are characterized by some heterogeneity between them, different inflammation locations and different types and levels of neuroinflammation [5]. In the last decade, several efforts were made to discover new biomarkers, allowing for a more accurate and rapid diagnosis for neurodegenerative diseases. These biomarkers include magnetic resonance imaging (MRI) whose targets are cerebral cortex, white matter, among others, positron emission tomography (PET) that essentially analyzes tau lesions and β-amyloid aggregations, cerebrospinal fluid (CSF) targeting β-amyloid peptides and tau peptides, blood biomarkers targeting also β-amyloid peptides, β-amyloid oligomers and tau peptides and genetic biomarkers [6,7]. Biomarkers aid in a more rapid and accurate diagnosis, providing indications in disease progression and in the identification of the best drug for a particular individual. The integration of these biomarkers into drug development or clinical trials for neurodegenerative diseases is an important step to help in the development and demonstration of drug efficacy and target engagement.

## 2. Alzheimer’s Disease

Alzheimer’s disease was firstly reported in 1906 by Alois Alzheimer and it is considered a slow, irreversible, complex and progressive disease [8,9].

AD represents between 50 and 75% of dementia cases, currently affecting 50 million people around the world and recent data and research show that in 2050, the prevalence of dementia and, consequently, of AD will increase two-times in Europe and three-times worldwide, reaching 113 million individuals [10,11]. Despite most AD symptoms beginning at 65 years old, it is thought that Alzheimer´s disease itself begins at 20 years old, long before the first symptoms [12].

AD symptoms include progressive memory loss, language and orientation difficulties, visuospatial functions, behavioral changes, cholinergic dysfunction, incapability of completing or conducting normal daily activities and, in the last case, dementia [13]. These symptoms result from the damage or destruction of the neurons involved in those actions [12]. Other symptoms of AD are correlated with neuropsychiatric symptoms (NPSs), which include delusions, hallucinations, depression, anxiety, irritability, sleep disorders, eating problems, aggressiveness and apathy. NPSs affect between 75 and 80% of AD patients [14,15].

Many causes are associated with AD, such as accumulation of extracellular β-amyloid peptides forming β-amyloid plaques, intracellular tau neurofibrillary tangles (NFTs) and brain atrophy [13,16,17]. It is also believed that the hippocampus is probably the first brain region affected in AD [18]. In the β-amyloid plaque case, they can interfere in the neuron–neuron communication at the synapses, contributing to the damage or death of neurons, which is called neurodegeneration. The cerebral accumulation of β-amyloid peptides (contains 36 to 43 amino acids) results from the unbalance between its production and removal in the brain [19,20]. Its formation derives from β abnormal cleavage of amyloid precursor protein (APP) [19]. The overexpression of APP also results in an increased formation of β-amyloid peptide, which is highly toxic, leading to AD [20].

NFTs have the power to block essential nutrients and other molecule transportation. This action contributes to the abnormal brain function and neuron non-survival.

Brain atrophy occurs because of the presence of toxic β-amyloid and tau phosphorylated proteins, which can activate microglia, the brain immune system capable of eliminating the toxic proteins and regulate homeostasis. When microglia are incapable of eliminating all toxic elements present in the brain, they may trigger pro-inflammatory or anti-inflammatory processes. A pro-inflammatory process promotes neuroinflammation through the secretion of cytokines and, consequently, leads to brain cell loss, resulting in brain atrophy [12,21].

AD has two different subtypes. The first one, a sporadic or late-onset AD (LOAD), that occurs after the age of 65, and a second one, early-onset AD (EOAD), is characterized by an earlier start, normally before 65 years of age [22].

EOAD is the most aggressive subtype of this disease, with an aggressive progression and lower life expectancy, which is nearly 100% heritable [23,24] K.J. Chang et al. [25] conducted a study where they found that EOAD patients have twice the odds of mortality risk than LOAD patients.

The principal cause of EOAD is the mutations in APP, presenilin 1(PSEN1) and presenilin 2 (PSEN2) genes. APP gene mutations are associated with 15% of EOAD cases, 5% with PSEN2 and almost 80% with PSEN1 [26].

In this subtype, normally, patients have more problems with visuospatial function, less motor coordination, lower attention capabilities, impaired memory and a higher cognitive dysfunction, leading to a faster functional degradation. The most affected brain areas of EOAD are the frontal, parietal and occipital lobes of the brain. However, EOAD patients have a lesser propensity for comorbidities, such as diabetes, obesity and circulatory diseases [27,28].

On the other hand, LOAD has a heterogeneous range with a lower heritability, with a maximum of 80% [24]. The major cause of LOAD is the genetic risk associated with the presence of allele ε4, which encodes the apolipoprotein E (ApoE) [26].

### 2.1. AD Risk Factors

The prevalence of Alzheimer’s disease is expected to rise in the next few years, especially because of the increasing life expectancy.

The greatest risk factors associated with AD are advanced age and genetics, among other risks.

#### 2.1.1. Age

Age is the principal risk factor associated with AD. The risk of developing AD increases 5.3% from 65 to 74 years, 13.8% at 75 years and more than 34.6% for those aged over 85 years old. Beyond age, women are more predisposed to have AD than men. Although age by itself is not responsible for the development of this disease and AD, it is not a normal symptom that comes with age [12,29].

#### 2.1.2. Genetics

In genetics, the ApoE-ε4 allele increases the risk of developing AD compared with ε2 and ε3 forms. ApoE protein is synthesized firstly in the liver, with a role in lipid binding, transportation through lymphatic and circulatory systems and cholesterol homeostasis.

ApoE is the most prevalent lipoprotein in the brain and, as described, plays a key role in cholesterol transport and recycling the lipids in the brain. Carrying at least one ApoE-ε4 allele increases the risk of AD by about three- or four-times, once this isoform is toxic to the brain. The ApoE-ε4 allele is found in 15–25% of the world’s general population and reaches over 50% in AD patients [30,31].

The APOE-ε4 allele affects the β-amyloid peptide, triggering its accumulation. NFT formation also affects microglia and the blood–brain barrier (BBB), causing its dysregulation that eventually contributes to neuronal stress and cell death and may accelerate DHA catabolism [32,33].

Coughlan and co-workers [34] developed a study where they found that APOE-ε4 interferes in the relation between DHA and entorhinal cortex and hippocampus volume. Further, APOE-ε4 and DHA have a significant interaction. This may imply that DHA disruption in BBB caused by APOE-ε4 can be related to AD.

On the other hand, APOE-ε2 has a protective response against AD.

Another genetic change is related with Down syndrome. Individuals with this disease have an increased risk of developing EOAD, where 50% of individuals with Down syndrome over 60 years probably develop AD [12].

Down syndrome is characterized by an extra copy of chromosome 21, which includes the gene that encodes the production of APP. APP is thought to play a key role in AD pathology once this gene in AD patients is cut and accumulated in β-amyloid plaques. Therefore, this extra copy may represent an increased possibility of AD development in Down syndrome patients. In the 1990s, two mutations in PSEN-1 and PSEN-2 were also described. These two genes located on chromosomes 14 and 1, respectively, are associated with EOAD cases, representing a few cases of AD disease [35].

#### 2.1.3. Other Risk Factors

Beyond age and genetic factors, family factors can also contribute to the development of AD. Individuals with first-degree relatives with Alzheimer’s have a higher possibility for AD disease. Other facts associated with lifestyle, such as smoking, obesity or diabetes, can contribute to AD symptoms. A recent study by Liguori et al. [36] showed that obstructive sleep apnea (OSA) may also be a risk factor for AD.

Diseases responsible for increasing the risk of cardiovascular disorders are directly associated with dementia. A healthy heart and vessels enable a healthy brain, ensuring that enough blood, oxygen and nutrients arrive to the brain [12]. It is estimated that cardiovascular diseases represent 50% of AD risk factors. Further, hypotension, mainly in late life, is associated with a higher risk of AD, while hypertension causes BBB dysfunction and increases vascular inflammation [37].

High cholesterol levels are also related to AD disease once the brain contains 25% of total cholesterol present in the human body and high-density lipoprotein cholesterol (HDL) is essential for synapse maintenance and maturation. Therefore, high cholesterol levels in the brain can cause malfunction of this cascade of events.

Studies also indicate that diabetes *mellitus* is related to AD through insulin inefficiency, although the mechanism is still unknown [38].

### 2.2. AD Treatments

Until today, many efforts have been made to discover an efficient treatment for an AD cure. However, none of the drugs available today for AD have the power to stop the damage and destruction of neurons and, consequently, to avoid that AD becomes a fatal disease.

The U.S Food and Drug Administration (FDA), until now, has approved seven drugs. Cholinesterase inhibitors are the first-line drugs in AD treatment and have proven to be efficacious for the cognitive and functional mild to moderate symptoms in AD disease [39]. Cholinesterases can also influence β-amyloid aggregation, due to the presence of a peripheral anionic site (PAS) in their structure [40,41].

Four of these drugs are tacrine (1993), which was withdrawn from the market in 2013 due to liver toxicity, donepezil (1996), rivastigmine (1998) and galantamine (2001) (Figure 1).

Acetylcholinesterase (AChE) and butyrylcholinesterase (BChE) are serine hydrolase enzymes that catalyze the hydrolysis of acetylcholine (ACh), present in mammalian brains.

AChE is located mainly in neurons and is essential in cholinergic neurotransmission, delivering ACh. BChE can be found in β-amyloid plaques and NFTs. It is known that it attenuates β-amyloid fibrillization in vitro, interacts with ApoE- ε4 and modulates the cholinergic activity and β-amyloid in the brain in vivo [42], although its role in AD disease has not yet been fully elucidated.

Macdonald et al. conducted a study with a mouse model with AD, where the BChE gene was knocked out and the absence of the BChE protein resulted in a decrease in β-amyloid deposition, suggesting that BChE can probably modulate β-amyloid formation [43].

There are findings suggesting that the inhibition of both AChE and BChE may have advantages, providing symptomatic benefits and slow disease progression [44,45]. Indeed, rivastigmine is a dual inhibitor of AChE and BChE enzymes, while donepezil and galantamine do not have a significant activity against BChE [39,40,41,42,43,44,45,46].

Memantine (2004), a partial N-methyl-D-aspartate (NMDA) receptor antagonist, and memantine combined with donepezil are also drugs approved to treat moderate to severe AD disease (Figure 1). The NMDA receptor antagonist allows one to prevent the effects of elevated glutamate levels, protecting against neurotoxicity and neural dysfunction [47] and reducing calcium influx into the cell.

These four drugs have adverse effects, such as dizziness, headache and confusion, and only help to provide a temporary relief of symptoms and pain. None of them can be considered as a disease modifier and they cannot reverse or slow down the progression of AD [8,12,30].

In June 2021, a new drug, aducanumab (Figure 2), was approved to treat AD, having, indeed, a disease-modifying potential [48]. It is an amyloid-targeting monoclonal antibody binding to β-amyloid amino acids 3 to 7, soluble oligomers, insoluble fibrils and amyloid β plaques, leading to a decrease in β-amyloid plaques in the AD brain (Figure 3) [48,49]. This mechanism takes place once this drug overpasses the blood–brain barrier.

Over the last few years, several new therapies for AD were tested but the majority failed in the clinical trial development phase. AD has, indeed, a diverse pathogenic etiology and the different major processes associated with this disease cover amyloid cascade, glutamatergic, oxidative stress, cholinergic, vascular and epigenetics hypotheses (Table 1) as well as other recent approaches, e.g., the inhibition of Fyn kinase activation by amyloid β-prion interaction and consequent inhibition of Tau hyperphosphorylation [50], identified as promising strategies for treatment against AD [47].

### 2.3. Natural Marine Sources Decreasing AD Symptoms

Environmental and lifestyle changes in the last few years have increased poor diets, which are related to chronic diseases, such as obesity, diabetes, cancer and neurodegenerative diseases.

In addition to health problems, world population ageing also continues to increase and, consequently, mental illnesses continue to rise. By considering these problems, it became important to identify new sources of neuroprotective drugs [69]. 

Many researchers found, in the marine environment, a huge source of unique and different structures, biologically and pharmacologically active [70]. Marine bioactive compounds have chemical properties not found in terrestrial products and have unique biological activities. Therefore, these resources are increasingly studied for drug discovery for several human diseases [71].

Marine ecosystems cover more than 70% of the Earth’s surface and contain around half of global diversity. The different conditions in the marine environment, such as temperature, pressure, salinity and light, allow marine organisms to produce a wide range of secondary metabolites, exhibiting a diversity of bioactivities, such as antioxidant, antimicrobial, anticancer, neuroprotective, antidiabetic and others [47,72].

The neuroprotective marine compounds are structurally very diverse, including polysaccharides, glycosaminoglycans, glycoproteins, lipids and glycolipids and pigments [71]. The marine resources producing active secondary metabolites are corals, sponges, algae, tunicates or marine bacteria [73,74]. Further, fish consumption is associated with decreased AD symptoms and incidence. Different fish species, such as mackerel, tuna and sardines, are a rich source of n−3 polyunsaturated fatty acids (PUFAs), especially docosahexaenoic acid (DHA) [75,76,77].

#### DHA

Docosahexaenoic acid (DHA, C22:6n−3) is an omega-3 long-chain polyunsaturated fatty acid. n−3 PUFAs are lipid components and can exist as triacylglycerols, phospholipids, free fatty acids and cholesterol esters (CEs), being classified by the number of carbon atoms and number and position of unsaturated bonds and playing an important role in cell membrane composition, cholesterol transport and energy storage. DHA is an important component in the phospholipid membrane, being the most abundant n−3 PUFA found in the brain and retina gray matter [78] and representing more than 30% (brain) and 90% (retina) of all n−3 PUFAs, respectively. Low concentrations of DHA in the brain are associated with several neurological diseases, including AD and Parkinson’s [79].

DHA can be obtained exogenously, especially from marine sources, such as fish oils, krill oil and algae, or endogenously through the bioconversion of essential α-linolenic acid (ALA) (Figure 4) [80]. The elongase and desaturase enzymes used to synthesize DHA are mostly expressed in the liver. ALA is desaturated by Δ6-desaturase in the endoplasmic reticulum (ER) and forms stearidonic acid, which is posteriorly desaturated by Δ5-desaturase into eicosapen-taenoic acid (EPA), which is elongated into docosapentaenoic acid (DPA) and then into 24:5n−3. 24:5n−3, suffering the action of Δ6-desaturase to form 24:6n−3, which is oxidized in DHA in peroxisome [81]. Welch et al. [82] referred to the fact that ALA conversion into EPA and DHA is variable between individuals. The limiting rate of production of EPA can reach 5% and DHA production only reaches 0.5% in the human body. These rates show that the inclusion of n−3 PUFA in the human diet is extremely important.

However, endogenous synthesis is low and becomes ineffective with ageing. Several clinical studies showed that increased intake of DHA can reduce the risk and delay the symptoms of AD [83]. 

Diverse studies demonstrated that DHA has many health benefits, e.g., it contributes to infant brain and eye development, prevents preterm birth, is essential for brain health and development, reduces the risk of tumors and some cancers, inhibits many inflammatory processes [84,85] and its consumption results in cardiovascular disease prevention. Its cardioprotective effects result from DHA’s capacity to modulate lipid metabolism, vascular function, cell membrane dynamics and from anti-inflammatory and antioxidant responses [55]. 

DHA is present at the phospholipids’ neuronal membrane and acts as a neuroprotective agent. It also has a role in neuronal deterioration, brain plasticity, synaptic loss and cholinergic transmission, improving neurogenesis and synaptogenesis. In a study carried out on mice, Debora Cutuli et al. [86] found that n−3 PUFA reduces apoptosis, neuronal density loss and glial degeneration. Further, DHA intake and pre-treatment clearly showed a neuroprotective and anti-inflammatory effect in the hippocampus once DHA beneficial effects are mostly concentrated in the hippocampus and entorhinal cortex. It reduces the prevalence of cognitive impairment and dysfunction of memory in AD, suppressing β-amyloid generation, acting as an antagonist by reducing the secretion of inflammatory mediators [78]. 

DHA can be obtained from triacylglycerols (TAGs) or phospholipids (PLs). However, as described by Ahn et al. [87] and Cardoso et al. [88], DHA-PLs show higher bioavailability and absorption than DHA-TAG. In the brain, DHA is esterified with PLs, namely phosphatidylethanolamine (PE), phosphatidylserine (PS), phosphatidylcholine (PC) (Figure 5) and others [89], but DHA-enriched phospholipids may also have lysophosphatidylcholine (DHA-LPC). DHA in marine organisms appears as DHA-PLs, namely as DHA-PC and DHA-PE, at sn−2 and sn−3 positions, respectively, of the glycerol moiety [90]. DHA-PLs can change their physical structure through environmental changing, altering cell response to permeability, absorption and carriage of phospholipid bilayer cell membrane [91].

Alongside DHA, EPA has a fundamental role in brain function as an anti-inflammatory and neuronal protective agent. EPA is also associated with lower incidence of AD, supporting the beneficial role of n−3 PUFA in mental health. Current data suggest an intake of 250 to 500 mg of EPA/DHA as daily doses. On the other hand, arachidonic acid (AA) is an n-6 PUFA and its prevalence over DHA promotes inflammation, contributing to neuronal diseases [92]. In 2017, Abdullah et al. [93] conducted a clinical trial with mouse models. In this study, they concluded that omega-3 fatty acid consumption is associated with lower AA/DHA and ApoE- ε4 ratios, contributing to a lower risk of developing AD.

## 3. DHA-Phospholipids from Marine Sources

In the last few years, DHA-PLs have attracted attention from researchers because of their unique health benefits. DHA-PLs are composed of phospholipids and DHA, phospholipids being responsible for structural and functional roles in cellular and subcellular membranes, for maintaining cellular barriers and are also precursors of lipid signaling molecules [94].

DHA-PLs have higher biological effects than other DHA forms and this combination also improves DHA oxidation stability. This can be explained by the fact that the aggregation between DHA and PLs increases the absorption rate and enables an easier passage of molecules through the intestinal wall. DHA-PLs have better bioavailability and higher content of n−3 PUFAs than lipids from the same source, allowing for a higher absorption, particularly in the brain, contributing to ameliorating several conditions as cardiovascular disease, anti-inflammatory effect, contributing to the improvement in cognitive ability, brain function and lipid metabolism [94]. DHA-PLs bioactivities cover anti-neurodegeneration, anti-cancer, neuroinflammation, alleviate attention deficit hyperactivity disorder (ADHD) symptoms and visual and obesity benefits [94,95,96,97,98,99].

DHA-PL-rich diets promote the release of acetylcholine, restore cholinergic activity, correct the value of PUFAs and balance the deterioration of the hippocampus, which is caused by ageing [94]. Specifically, Che and coworkers [100] demonstrated that DHA-PC inhibits the generation and accumulation of β-amyloid, one of AD’s causes. Another study, by Min et al. [101], in rat models, demonstrated that DHA-PC and DHA-PS have a neuroprotective effect, improving learning and memory ability in dementia in aged rats. They also inhibit tau phosphorylation, suppressing neuroinflammation. Further, Zhou et al. [102] obtained results in rats suggesting that DHA-PC and DHA-PS can improve oxidative stress in the hippocampus and have benefits on mitochondrial damage and DHA-PS also increases the production of insoluble β-amyloid in AD.

DHA-PLs are also more effective and functional in DHA transportation, lipid absorption and metabolism, causing better regulation in obesity conditions and multiple sclerosis treatment over DHA-TAG.

PLs can be extracted through organic solvents, atmospheric oxygen and high temperature and supercritical carbon dioxide (SC-CO_2_). SC-CO_2_ technology is the most effective method, allowing for a better final product quality, being environmentally friendly, providing a higher purity and yield and faster extraction period than other extraction methodologies [103].

DHA-PLs can be extracted from marine organisms (fish and krill), mollusks and some by-products of marine biological processing. Phospholipids obtained from marine organisms are recommended for food, pharmaceutical and cosmetic industries, mainly due to their amphiphilic character [94].

In the next sections, the known and the new marine sources rich in DHA-PLs are revised, showing the importance of their use in the human diet. Although algae of marine origin contain phospholipids, these are not DHA-rich phospholipids and, therefore, they are not within the scope of the present review.

### 3.1. DHA-PLs in Fish

Fish is one of the most important sources of DHA-PLs. Several species, such as salmon, chub mackerel, Atlantic herring, boarfish and sardines, are rich in DHA-PLs. The DHA-PL composition and content vary with their growth environment, feeding and pressure differences.

Farkas et al. [104] compared the phospholipid composition in the brain of some species with different water temperatures. The results suggested that cold-blooded species’ brains are richer in DHA than warm-blooded species. This fact supports some evidence that fish brain DHA increases with cold temperature [104].

Already in 1997, Bandarra et al. targeted [77] sardine (*Sardina pilchardus)* lipid composition analysis. In this study, the values of PL class were lower than TAG values for lean and fatty sardines. However, when the fatty acid profile was evaluated for neutral lipids (TAG included) and polar lipids (PL included), the PL fraction showed a much higher amount of DHA, proving the richness of this species in DHA-PL [77].

The lipid profile of young Atlantic salmon (*Salmo salar* Linnaeus 1758.) was studied by Nefedova et al. [105]. A comparative analysis of the lipid profile of young salmon showed that PL is the prevalent lipid class. Until the end of the first year of life, salmon has a high composition (%) of DHA, but with ageing, the percentage of DHA decreases.

Another three species were studied by Egerton et al. [106], namely blue whiting (*Micromesistius poutassou*), boarfish (*Capros aper*) and Atlantic herring (*Clupea harengus*). In this work, they analyzed TAG fatty acid composition and PL fatty acid composition, separately. Comparing the two analyses, the PL class comprises the largest fraction of PUFA and, consequently, the highest percentage of DHA in all species, being more than twice as high. Regarding the three species, DHA is the most abundant fatty acid in all three and herring showed the highest relative concentration of DHA (35.38%). Herring also contains the larger relative concentration of PUFA.

In 2020, a study was conducted with chub mackerel (*Scomber colias).* Despite TAG representing more than 80% of the total lipid content in all the seasons, the PL fraction possesses the highest percentages of DHA. This fact also contributes to the highest percentage of n−3 PUFA in the PL fraction [76].

### 3.2. DHA-PLs in Krill

Antarctic krill (*Euphausia superba*), a shrimp with one of the largest biomasses in multicellular species, attracted attention due to its unique composition and health benefits for humans. This species is abundant in n−3 PUFA, especially DHA, which is manly esterified/linked to *sn*-2 position of PL rather than linked in TAG. In krill oil, more than 40% of total lipids are PL and it is known that DHA-PL is more bioavailable than DHA-TAG [107,108].

Zhou et al. [109] conducted a study where they analyzed the molecular species presented in krill oil. DHA-PL amount was higher than AA amount, which is pro-inflammatory, proving why krill oil plays a key role in anti-inflammatory processes. Among the DHA-PL forms, the most abundant was DHA-PC with 8676.65 nmol/g, as also confirmed by Showman and coworkers [110].

Sistilli and coworkers [111] found that DHA-PLs are more effective in reducing hepatic steatosis than DHA-TAG. The treatment with DHA-PL reduced more than 40% of liver steatosis symptoms. DHA-PL from krill oil also demonstrated a strong impact in obesity treatment, exhibiting antisteatotic bioactivity [112].

In 2022, novel research was developed by Sung et al. [97], the main goal of which was to compare the impact in plasma of a 30-day supplementation with krill oil and fish oil. They concluded that krill oil treatment increased significantly in 45% DHA- PLs in plasma compared with fish oil treatment. These results also corroborate PL higher bioavailability, rather than TAG.

Another study comparing krill oil and fish oil was developed by Ahn et al. [72]. They showed, using chromatographic methods, that krill oil has a high PL content. DHA-PL amount remaining in the blood and brain after ingestion was higher than with fish oil ingestion, possibly due to PL chemical structure in krill oil. These facts suggest that krill oil has potential for neurodegenerative and cardiovascular diseases [113,114]. In 2008, Zhu et al. [115] also proved the benefits of dietary krill oil in colon cancer treatment.

Another important study was conducted by Li et al. [116] with the investigation of the role of Antarctic krill oil against AD. The mice treated with krill oil showed a significant amelioration in learning and memory deficits and this research also indicated that krill oil can reduce β-amyloid accumulation in the hippocampus.

Krill oil has several relevant bioactivities, namely anti-inflammatory, anti-obesity and antidiabetic, neuroprotective and anticancer. Nowadays, krill oil is available on the market as soft gels, capsules, gummies and tablets [117].

### 3.3. DHA-PLs in Mollusks

Mollusks are one of the most diverse groups, representing around 23% of all marine organisms. Phylum Mollusca has attracted high interest for its secondary metabolites, which are antitumor, anti-inflammatory and antimicrobial [118].

Zhukova studied the fatty acid composition of sixteen marine mollusks from the East Pacific [119]. N-3 PUFA accounts for the majority of total fatty acids and DHA, presenting the highest content in almost all species. Indeed, the DHA ratio derives from these mollusk diets based on zooplankton, diatoms and dinoflagellates, once DHA prevails in zooplankton and dinoflagellate fatty acid composition [119]. The n−3 and n-6 PUFA families are essential for important physiological processes. Their major source is marine ecosystems. The fatty acids (FAs) from phytoplankton, which are the primary producer of organic matter and PUFAs, are transferred into consumers via food webs. Mollusk FAs have attracted the attention of researchers that has been driven by their critical roles in aquatic ecology and their importance as sources of essential PUFAs. The main objective of this review is to focus on the most important factors and causes determining the biodiversity of the mollusk FAs, with an emphasis on the key relationship of these FAs with the food spectrum and trophic preference. The marker FAs of trophic sources are also of particular interest. The discovery of new symbioses involving invertebrates and bacteria, which are responsible for nutrition of the host, deserves special attention. The present paper also highlights recent research into the molecular and biochemical mechanisms of PUFA biosynthesis in marine mollusks. The biosynthetic capacities of marine mollusks require a well-grounded evaluation [119].

Another study was developed by Tabakaeva and Tabakaev [120] with *Spisula sachalinensis*, a bivalve mollusk. They analyzed their lipid classes and found that PL was the main class in all soft tissues (mantle, adductor, locomotor muscle and internals) of this species, with internals having the highest percentage (45.6%). Regarding fatty acid composition, DHA is the major PUFA component, with its percentage being higher in adductor, which is common in this type of mollusks.

Krishnan and coworkers [121] studied the nutritional profile of several edible marine mollusks, including Indian squid (*Uroteuthis (Photololigo duvaucelii)*, octopus (*Amphioctopus marginatus*), cuttlefish (*Sepiella inermis*) and oyster (*Crassostrea. bilineata*)). Between them, *Amphioctopus marginatus* has the highest percentage of DHA and *Crassostrea bilineata* the lowest. Despite these mollusks showing lower lipid content in relation to nudibranchs mollusks, they have much higher percentages of n−3 PUFA.

In 2019, Kapranova et al. [122] developed the first study on the mussel *Mytilus galloprovincialis Lam*. (1819), where they evaluated the fatty acid composition of eggs and sperm. The main n−3 PUFA in mussel sperm and eggs is EPA and DHA.

### 3.4. DHA-PLs in Marine By-Products

Fisheries and aquaculture are one of the most important industries for human feed. In total, only 50–60% of fishery is used for human consumption. The rest is considered undesired and called by-products [123]. The main by-products are heads, blood, viscera, skin and tails, which have the highest concentration of high-added-value compounds, including lipids, proteins, minerals and vitamins [124].

Marine by-products are the newest source of DHA-PLs, contributing to concerns in recent years, such as zero waste and circular economy, to promote a sustainable world. 

One of these lipid by-products is based on the Patagonian squid (*Doriteuthis gahi*), which is abundant along the pacific coast of South America. A study by Aubourg et al. [123] in 2021 showed high values of PLs, mostly in winter (463.5 g/kg), while DHA was the most abundant PUFA (30.79/100 g). The results obtained by this group reinforce the importance and value of by-products as sources for nutraceutical, health, pharmaceutical and cosmeceutical industries.

Further, the non-edible parts of crustaceans can be considered rich and valuable bioactive compounds. In 2021, a study by Zhang et al. [125] in *Penaeus Vannamei* head using lipidomics confirmed the relevance of this body part of shrimp, with a high level of phospholipids. The lipid fraction is also known for its positive effect on cardiovascular disease [125]. However, nearly 60 wt% (weight percentage) of shrimp body mass is wasted [126].

*Parapenaeus longirostris* is a shrimp species mostly found in the Mediterranean. Messina et al. [127] developed a study with *Parapenaeus longirostris* by-products, specifically in the exoskeleton. Through its fatty acid profile analyses, it was possible to conclude that DHA is the fatty acid appearing with the highest content amongst the n−3 PUFAs. These findings support the possibility of using these by-products, once they are rich in DHA, for a possible treatment or symptom relief for neurodegenerative diseases, such as AD.

Ahmad and coworkers [128] investigated Australian seafood by-products, namely from octopus (*Octopus tetricus*), squid (*Sepioteuthis australis*), sardine (*Sardinops sagax*), salmon (*Salmo salar*) and prawn (*Penaeus plebejus*). In this work, they studied the fatty acid profiles of viscera and head. The results showed that all species are rich in DHA. The by-products contain more quantity per 100 g of DHA than flesh (edible part) in all species, except in squid. Further, Liu et al. [129] studied phospholipid extracts from by-products of codfish roe, squid gonad and shrimp head. In this work, the three species showed high quantities of phospholipids with the highest values shown in PC (53.6% for shrimp head) and PE (31.2% for codfish roe). These marine PLs have a high content of n−3 PUFAs, binding predominantly to DHA, DPA and EPA, which reinforces their anti-atherosclerotic property.

Ahmmed et al. [130] studied the phospholipid content and fatty acid composition of four by-products (blue mackerel roe, head, skin and male gonad) of pacific blue mackerel (*Scomber australasicus*). All analyzed by-products had a considerable content of PLs, but blue mackerel roe demonstrated to be the best source of PLs (38.6 µmol/g), representing PC 64.5% of total PLs.

Among fatty acid composition, DHA was the most representative one in the four by-products. Blue mackerel roe, with a total of 27.5%, has the highest percentage of DHA and skin the lowest (19.5%).

Interestingly, all these bioactives present in by-products can be extracted using green methods and contribute to reaching the zero-waste goal through finding applications in nutraceuticals and pharmaceuticals. They should, indeed, be implemented within a circular economy approach for a sustainable and healthy future.

In Table 2, the sources of DHA-PLs, their concentration and classes are summarized.

## 4. Conclusions

This review focuses on DHA, which has, indeed, unique health benefits, particularly for Alzheimer’s disease and cardiovascular disease prevention, and as an inhibitor of inflammatory processes. This n−3 PUFA is present in the phospholipid neuronal membrane and is the main n−3 PUFA present in the brain, suppressing β-amyloid production and reducing inflammatory mediators. Some studies showed that DHA-PLs have a higher bioavailability and absorption in the brain than that of DHA-TAG, contributing to the prevention of AD, a multifactorial fatal disease. Hypotheses for AD treatment are highlighted, namely the amyloid cascade, epigenetics or cholinergic strategies, which can contribute to identify promising treatments and the current drugs used for symptomatic treatment are presented.

In recent years, the scientific community has identified marine sources of DHA but this review focuses on the latest findings on the richest sources of DHA-PLs, namely the research carried out in sardine, chub mackerel or Atlantic herring, showing that DHA is present in higher amounts in PLs. Further, in krill oil, 40% of total lipids are PLs rich in DHA. These bioactives reduce hepatic steatosis and their benefits for colon cancer treatment have been identified.

Marine by-products are the non-edible part of fisheries and aquaculture products not used for human consumption and represent around 40% of the total products. In some species, the by-product may contain a higher percentage of DHA than edible parts.

Some clinical trials were already developed with DHA-phospholipid products. One of these studies was by Rondanelli et al. [131], showing that food supplements composed of 720 mg of DHA, 286 mg of EPA, 16 mg of vitamin E, 160 mg of soy phospholipids, 95 mg of tryptophan and 5 mg of melatonin, given for 3 months, significantly improved cognitive impairment in the long term in elderly people. Although it did not occur with DHA-PLs only and with marine sources, this clinical trial shows the benefits of DHA-PL intake.

It is important to highlight the fact that all these by-products can be extracted using green methods and contribute to circular economy and the zero-waste goal. Indeed, this review shows the importance of marine sector zero waste in the framework of a circular economy, promoting a healthy sustainable future.

## Figures and Tables

**Figure 1 marinedrugs-20-00662-f001:**
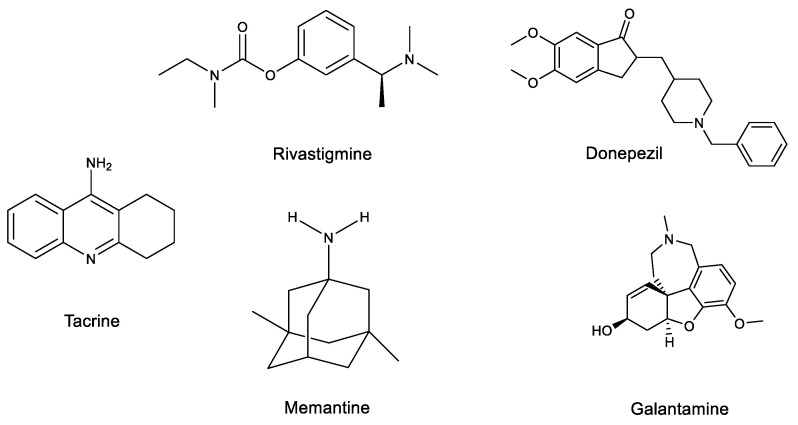
Chemical structure of tacrine, rivastigmine, donepezil, memantine and galantamine.

**Figure 2 marinedrugs-20-00662-f002:**
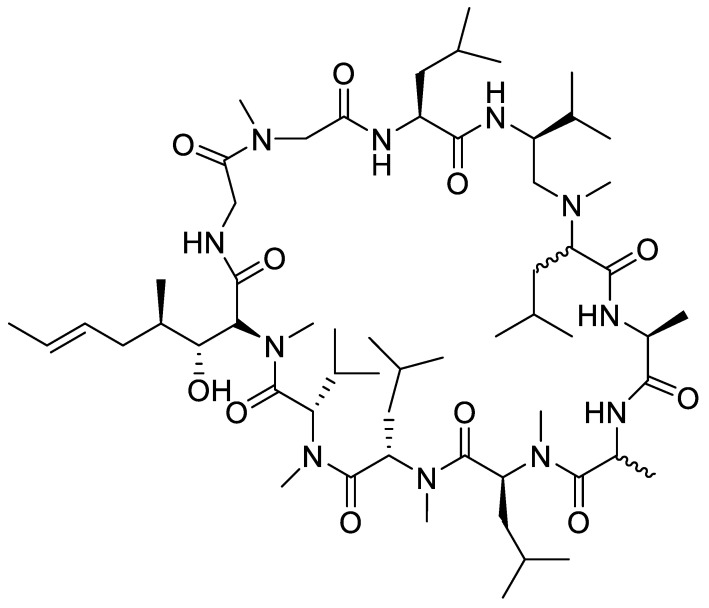
Aducanumab chemical structure.

**Figure 3 marinedrugs-20-00662-f003:**
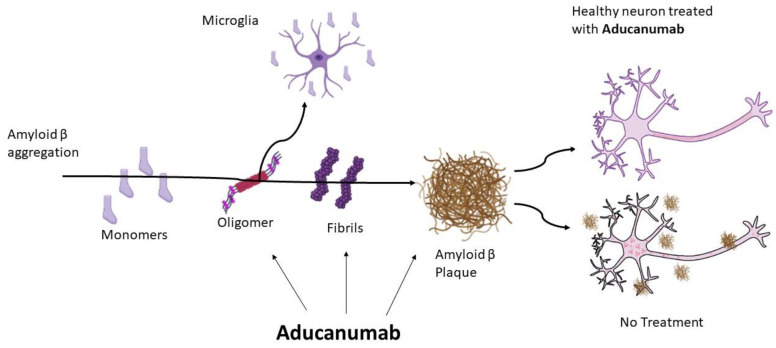
Aducanumab mode of action, targeting β-amyloid plaque formation, by binding soluble oligomers and insoluble fibrils, precursors of amyloid β plaques in the brain. Aducanumab has a high selectivity allowing for the aggregation of amyloid β forms but reducing amyloid β plaques in the brain, as shown in the healthy neuron after treatment with this drug.

**Figure 4 marinedrugs-20-00662-f004:**
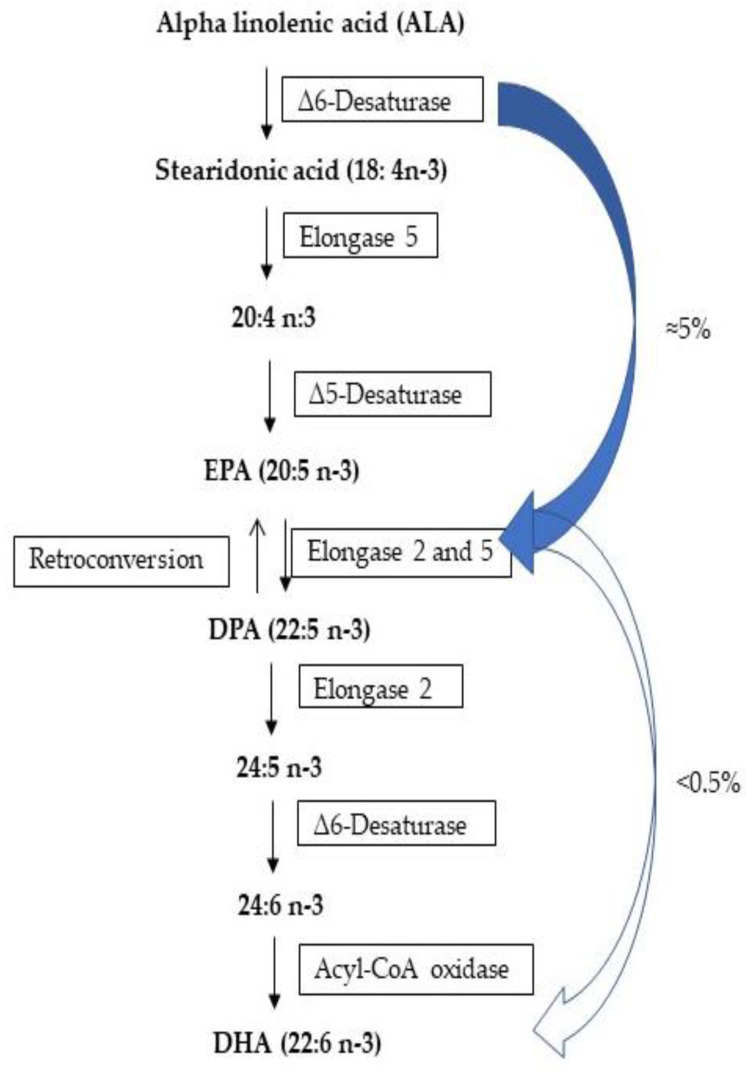
Bioconversion of ALA in DHA [81].

**Figure 5 marinedrugs-20-00662-f005:**
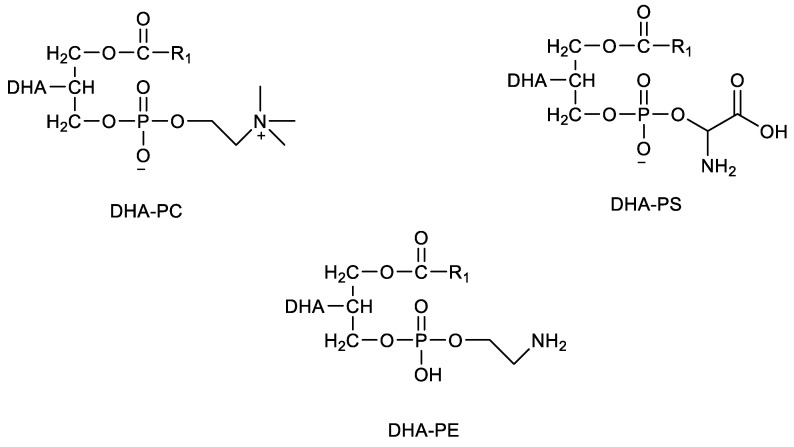
Chemical structure of DHA-PC, DHA-PE and DHA-PS; R_1_—saturated fatty acid.

**Table 1 marinedrugs-20-00662-t001:** Hypotheses considered for AD drug development.

Hypothesis	Explanation	References
Amyloid cascade	Amyloid hypothesis was developed in 1992. Amyloid cascade hypothesis is associated with the presence of soluble toxic oligomers from β-amyloid. Monomers of β-amyloid are produced through life however changes in amyloid cascade result from the unbalance between the production and the clearance of β-amyloid peptides resulting in its accumulation and aggregation in the brain. The consequences of this hypothesis are inflammation, oxidative stress, neuronal injury and death and tau pathology.	[51,52,53,54]
Vascular	This hypothesis was developed in 1993. Several vascular risk factors, such as atherosclerosis, hypertension, diabetes, or heart diseases are related with the increasing of β-amyloid production, dysregulation of BBB, decreased cerebral blood flow, tau hyperphosphorylation. Vascular cell dysfunction affects BBB permeability messing with the regulation of brain metabolites, accelerating β-amyloid pathology. These events cause the reduction of β-amyloid clearance, promoting the production and aggregation of β-amyloid, that leads to a lower neuronal activity and progressive neurodegeneration and neuronal death.	[38,55]
Epigenetics	Recently, studies regarding epigenetics (study of gene expression modification and/or chromatin structure) suggest that this hypothesis may contribute to the risk of AD development. DHA methylation and histone acetylation are responsible for gene expression at the transcriptional level. DHA methylation is catalyzed by DHA methyl transferases (DNMTs) with DNMT1 being related to faulty DHA methylation leading dementia and to neurodegeneration.	[29,56,57]
Oxidative stress (OS)	The brain is the human structure that uses more oxygen. OS can be defined as the imbalance (incapacity of cells to remove) of antioxidant system between reactive oxygen species (ROS) and/or reactive nitrogen species (RNS). The decreased levels of antioxidants and excessive generation of ROS and/or RNS can lead to macromolecular damage, oxidative damage to cells (loss of function and apoptosis), molecules and biological systems.OS is considered an early event in AD.	[58,59,60,61]
Glutamatergic	Over 40% of neuronal synapses are glutamatergic and glutamate level is regulated by metabolite swap between neuronal, astrocytic and endothelial cells. So, β-amyloid binding to glutamate receptors provokes the overactivation of glutamate receptors causing glutamate accumulation and resulting in glutamatergic synapse loss and synapse dysfunction, leading to neuronal swelling, destruction of membrane integrity and cell death, which is related to AD.	[62,63,64,65]
Cholinergic	Cholinergic neurons use acetylcholine (ACh) as neurotransmitter, which is directly correlated with physiological processes. So, normal cognitive function depends on a correct cholinergic neurotransmission. ACh is essential for brain especially in memory functions. So, decrease of ACh levels, due to decrease of the enzyme that synthesizes ACh, results in cholinergic loss in cortical structures, such as cerebral cortex, basal forebrain and hippocampus, that are associated with AD. Inhibitors of AChE and of BChE increase ACh levels in the brain.	[66,67,68]

**Table 2 marinedrugs-20-00662-t002:** Sources of DHA-PLs and their concentration and classes.

Sources	Tissues/Organs	DHA-PLs Concentration	PLs Classes	References
Fish				
*Sardina pilchardus*	Edible part	3.65–11.25%	PC, PE	[77]
*Salmo salar* L.	Edible part	3.95 ± 0.56 to 8.12 ± 1.45%	PS, PC, LPC	[105]
*Micromesistius poutassou* ^1^	Edible part	24.63 ± 2.19%	-	[106]
*Capros aper* ^1^	Edible part	29.35 ± 2.24%	-	[106]
*Clupea harengus* ^1^	Edible part	35.38 ± 4.75%	-	[106]
*Scomber colias* ^1^	Edible part	4.4 ± 0.2 to 8.4 ± 0.9%	-	[76]
Krill				
*Euphausia superba* ^1^	Muscle	54.0%	-	[102]
*Mollusks*				
*Spisula sachalinensis* ^1^	Mantle, Adductor, locomotor muscle, internals	40.3 ± 2.02% 35.0 ±1.70% 37.5 ± 1.91% 45.6 ± 2.22%	-	[120]
*By-products*				
*Doriteuthis gahi*	Viscera, skin, cartilage	359.2 to 463.5 (g/kg)	PC, PE	[124]
*Penaeus Vannamei*	Head	45.7%	PC, PE, PS	[124]
*Scomber australasicus*	roe, head, skin, male gonad	37.6 ± 2.1% 21.7 ± 1.1% 13.4 ± 1.1% 26.9 ± 1.4%	PC, PE, LPC	[124]
Codfish	Roe	58.92 ± 1.36%	PC, PE, LPC	[124]
Squid	Gonad	69.71 ± 0.55%	PC, PE, LPC	[124]
Shrimp	Head	66.44 ± 0.81%	PC, PE, LPC	[124]

^1^ In these species PL classes are not specified, andthe values in DHA-PLs concentration are referent to the total of PLs.

## Data Availability

Not applicable.

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
