# Peer review of "Marine Sources of DHA-Rich Phospholipids with Anti-Alzheimer Effect"

_marinedrugs, 2022, doi:10.3390/md20110662_

Round 1

Reviewer 1 Report

Manuscript: New Sources of DHA-rich Phospholipids with Anti-Alzheimer Effect

This is a very interesting manuscript, well written, that provides abundant background to the subject, such as the review of the causes of Alzheimer's and its treatments.

- Regarding the title, I suggest removing the term "New" since many sources of traditional phospholipids are reviewed, such as fish.

- The description of sources of phospholipids of algal origin, which are abundant, is missing.

- It would be necessary to improve the scope and objectives of the manuscript to provide quantification data. I suggest the authors make a table with the sources of phospholipids, including algae, and provide concentration data in tissues/organs/organisms.

- The classes of phospholipids contained in each source should be indicated in this table.

- There are many gaps in the literature. All previous work on phospholipids and their significance in Alzheimer's should be discussed. For example, look at:

- Kosicek, M., & Hecimovic, S. (2013). Phospholipids and Alzheimer's disease: alterations, mechanisms and potential biomarkers. International journal of molecular sciences, 14(1), 1310-1322.

- Farooqui, A.A., Liss, L., & Horrocks, L.A. (1988). Neurochemical aspects of Alzheimer's disease: involvement of membrane phospholipids. Metabolic brain disease, 3(1), 19-35.

- Wells, K., Farooqui, A. A., Liss, L., & Horrocks, L. A. (1995). Neural membrane phospholipids in Alzheimer disease. Neurochemical research, 20(11), 1329-1333.

- Pettegrew, J. W., Panchalingam, K., Hamilton, R. L., & McClure, R. J. (2001). Brain membrane phospholipid alterations in Alzheimer's disease. Neurochemical research, 26(7), 771-782.

These works cited are classics on this topic. Other recent works on this topic should be especially considered. For instance, look at:

- Ojo, J. O., Algamal, M., Leary, P., Abdullah, L., Mouzon, B., Evans, J. E., ... & Crawford, F. (2019). Converging and differential brain phospholipid dysregulation in the pathogenesis of repetitive mild traumatic brain injury and Alzheimer's disease. Frontiers in Neuroscience, 13, 103.

- Scollo, F., Tempra, C., Lolicato, F., Sciacca, M. F., Raudino, A., Milardi, D., & La Rosa, C. (2018). Phospholipids critical micellar concentrations trigger different mechanisms of intrinsically disordered proteins interaction with model membranes. The journal of physical chemistry letters, 9(17), 5125-5129.

On the other hand, the manuscript should consider discussing the interactions of DHA with other PUFAs, such as EPA and ARA, in terms of their significance in this pathology. Look:

Abdullah, L., Evans, J. E., Emmerich, T., Crynen, G., Shackleton, B., Keegan, A. P., ... & Bachmeier, C. (2017). APOE ε4 specific imbalance of arachidonic acid and docosahexaenoic acid in serum phospholipids identifies individuals with preclinical Mild Cognitive Impairment/Alzheimer's Disease. Aging (Albany NY), 9(3), 964.

In summary, this is a promising manuscript, missing some sections, and providing interesting data on phospholipid sources.

Author Response

Dear Reviewer 1,

We thank you so much for your comments and we are pleased to upload all changes made according to your comments. Pleas be so kind to see the attachement.

With my best regards

Amelia Pilar Rauter

Reviewer 2 Report

The manuscript by Ferreira Ines et al. reviews the DHA marine sources, focusing on the possible health benefits of the DHA-rich phospholipids, and their potential for future AD treatment. Overall, it is an interesting manuscript, considering an interesting issue which is the natural compounds based on possible future AD therapy. The bibliography is very well prepared, based on the latest scientific reports. The work is appropriate for this Journal, however in my opinion some issues need to be completed, and subsequently, a minor revision is required. Some paragraphs need to be corrected, the tables and figures should be revised, and there are also some suggestions for the Authors to make the manuscript more adequate to its title. The text needs to be revised carefully as there are some typological and punctuation errors.

The most important comments:

Abstract:

The Authors need to explain the PUFA abbreviation in the abstract.

Section 1: Neurodegenerative Diseases

Consider exchanging the word “symptom” for “characteristic” in line 36.

In the last paragraph, the biomarkers are mentioned, and no information is provided about their role in neurodegenerative diseases. It is pointed out in the next paragraph, however – one simple sentence would be valuable here for better understanding.

Section 2: Alzheimer’s Disease

Line 68 – at least one more citation is required.

The paragraph starting from line 77 – the Authors should revise this one. The phrases “this accumulation” (line 82) and “this protein” are not clear to the reader – which protein accumulates?

Also – the sentence describing the APP (line 83) is not correct – APP is not located in the chromosome but is the gene encoding it. Please rewrite the sentence and make it more clear for the reader. The overexpression of APP is not because the APP encoding gene is located on the 21 chromosome but it’s observed only in case of the Down trisomy 21. Some explanation is missing, or maybe even better to omit chromosome 21 here as it’s well described in the text later.

Section 2.1.2: Genetics

Kindly revise the spelling inconsistency of the ApoE-ε4 allele (with “-“ and without it) and Down syndrome (capital letter in line 142 and small letter in line 143).

The PEN-1 and PEN-2 abbreviations are inconsistent with previously explained ones (PSEN1 and PSEN2).

Section 2.2. AD treatments

The Authors do not mention tacrine which was the first FDA-approved AD drug in 1993 (no matter if it was withdrawn later from the market). Consider adding it.

Also, in my opinion, in this section, the Authors need to add some information about butyrylcholinesterase and its role in AD brains, especially since rivastigmine is considered a dual AChE and BuChe inhibitor.

Figure 1 should be revised and corrected – kindly clean up the structures (different sizes of structures, in case of memantine not the best presentation of the molecule).

In the paragraph starting from line 187 – The Authors claim the cholinesterase inhibitors cannot modify the curse of AD. In fact – to some extent of course – they can. Kindly please note the possible results of inhibition of cholinesterase by inhibitors binding to its peripheral active site (PAS). Kindly consider changing slightly.

Line 207 – this sentence is not clear to me, the part starting from “e.g.”

Table 1 – last line – describing cholinergic hypothesis of AD – Authors do not mention the most important role of cholinesterases in the cholinergic dysfunction – AChE and BuChE enzymes need to be highlighted here.

Section 2.3.1. DHA

Figure 4,5 – there’s no mention of these ones in the text - in my opinion it’s better to refer in the text to all the figures and tables.

Also – Figure 4 itself is not fully clear – what do the 5% and 0.5% values mean?

Figure 5 – in my opinion, there’s no need to present these structures – kindly consider exchanging them for the DHA-PLs structures.

Section 3. DHA-PLs from marine sources

The major correction needed to be done is to highlight the NEW sources of DHA (as the Authors claim in the title of the manuscript). Try to pay attention to new interesting marine sources in comparison to the already well-known ones and also mark in text from time to time, that new sources are described.

Line 332/333 – “in the next sections […] marine NEW sources are revised […]

Figure 6  – there’s no mention of this one in the text – also, consider presenting the structures of PLs in Figure 5 only and omitting Figure 6.

Section 3.3 DHA-PLs in mollusks

Line 407 – there’s no need to explain the PUFA abbreviation here, it was already explained earlier in the text.

Line 419/420 – Sentence not clear – try to rewrite.

Line 428-429 – Kindly don’t use the abbreviations in the Latin names of mollusks

Section 3.4. DHA-PLs in marine by-products

Many times, in this section Authors use the abbreviation “et al.” with a normal font, while earlier in the previous sections it was written in Italics. Please make this consistent throughout the whole text.

Line 452 – the “wt%” explanation is missing.

Section 4. Conclusions

It would be valuable to add some sentences about the clinical trials with DHA-rich products if any were/are conducted.

Author Response

Dear reviewer 2,

Thank you very much for your comments. Please be so kind to see the attachement, in which we identify all changes made in the manuscript, according to your comments.

Round 2

Reviewer 1 Report

The suggestions have been satisfactorily resolved, so I consider that the manuscript is appropriate to be published in its current form.